# Structural, Thermal and Magnetic Analysis of Fe_75_Co_10_Nb_6_B_9_ and Fe_65_Co_20_Nb_6_B_9_ Nanostructured Alloys

**DOI:** 10.3390/ma14164542

**Published:** 2021-08-12

**Authors:** Albert Carrillo, Jason Daza, Joan Saurina, Lluisa Escoda, Joan-Josep Suñol

**Affiliations:** Department of Physics, Higher Polytechnic School, Campus Montilivi s/n, University of Girona, 17003 Girona, Spain; albert.carrillo.berlanga@gmail.com (A.C.); jason0@gmail.com (J.D.); joan.saurina@udg.edu (J.S.); lluisa.escoda@udg.edu (L.E.)

**Keywords:** mechanical alloying, Fe-Co-Nb-B alloys, ferromagnetic alloys, thermal analysis, X-ray diffraction

## Abstract

Two nanocrystalline ferromagnetic alloys of the Fe-Co-Nb-B system have been produced by mechanical alloying (MA). Their microstructure, thermal behavior and magnetic response were checked by X-ray diffraction (XRD), differential scanning calorimetry (DSC) and vibrating sample magnetometry (VSM). After 80 h of MA, the alloys were nanostructured (bcc-Fe(Co)-rich phase). As the Co content increases, the density of the dislocations decreases. Besides, a higher concentration of Co causes an increase in the activation energy of the crystallization process. The calculated energies, 267 and 332 kJ/mol, are associated to the crystalline growth of the bcc-Fe-rich phase. The Co content of the samples has no effect on the value of the saturation magnetization, whereas the coercivity is lower in the alloy containing less Co. Samples were compacted and heat-treated. Optimal annealing reduces the coercivity by a factor of two. Results were compared with the data of Fe-Nb-B and Fe-Ni-Nb-B alloys.

## 1. Introduction

In recent decades, interest in the development of nanocrystalline soft ferromagnetic materials has increased [1]. Soft and semi-hard magnetic materials are characterized by low-medium coercivity and high saturation magnetization. Among the multiple systems that have been analyzed, we should remark the Finemet, Nanoperm and Hitperm alloys, with elemental compositions Fe-M-Si-B-Cu, Fe-M-B-Cu and Fe-Co-M-B-Cu, respectively (M is an early transition metal) [2,3,4]. Thus, Hitperm alloys are characterized by the addition of Co and Fe as ferromagnetic metals, an early transition metal, such as Nb, Zr or Hf, a metalloid, such as B, P or Si, and a late transition metal, which is usually Cu. Mechanical alloying (MA) is a technique that allows to obtain nanocrystalline alloys directly in powder form [5,6], which can be compacted as a powder metallurgical route [7].

The addition of Co serves two purposes. On the one hand, it helps to maximize the magnetic moment (increasing the saturation magnetization) [8] because Fe-Co alloys are at the top of the Slater–Pauling curve. On the other hand, Co increases the Curie temperature (>1000 K) of both the residual disordered grain boundaries and the nanocrystals [9]. Therefore, Hitperm alloys were designed as high-temperature soft magnetic materials (preventing grains from decoupling magnetically) and allows for the operation of these alloys at much larger temperatures than the other families. However, for high temperature applications, the challenge is not only connected to the Curie temperature of the phases; since we are dealing with metastable materials, we also have to take into account the temperature at which microstructural transformation will take place. This family of alloys fulfills both requirements. In addition, minor amounts of Co increase the magnetic saturation while inhibiting the coercivity increase [10]. Nevertheless, a high amount of Co diminishes the softness (in alloys with Cu) by increasing both the grain size and the coercivity [11]. It is also known that the presence of Nb instead of Zr or Hf reduces the crystalline size, which softens the alloys according to the random anisotropy model [12]. The addition of Cu and Nb favors the thermal stabilization of the nanocrystalline phase from crystalline growth [13]. Cu-free Hitperm alloys possess lower values of coercivity compared to Cu containing Hitperm [14] because some Cu atoms are on a solid solution in the substitutional sites. The applications of these materials include sensors, core transformers, thin film inductors, magnetic flux amplifiers, magnetocaloric systems [15,16] and micro-devices for avionic [17].

In this work, two Cu-free Hitperm-type alloys have been produced, and their thermal, structural and magnetic characterization have been carried out. It is interesting to produce alloys with a low coercivity and high saturation magnetization. Furthermore, the compaction of the powdered alloys allows to obtain complex bulk shapes that cannot be obtained by means of rapid solidification techniques.

## 2. Experimental Details

Two alloys (Fe_75_Co_10_Nb_6_B_9_ and Fe_65_Co_20_Nb_6_B_9_ in at.%, labelled as Co10 and Co20, respectively) were produced by ball milling in a planetary mill: a Fritsch Pulverisette 7 (Fritsch, Idar-Oberstein, Germany). The powdered elements Fe, Co, Nb and B (Sigma Aldrich, Saint Louis, MO, USA) have a high purity (<99.5%) and a low particle size (<100 µm). The powders were mixed in order to obtain the nominal atomic composition. Moreover, the milling atmosphere was inert (powders and balls were sealed with Teflon in a glove box after three controlled vacuum–argon processes). The material of the balls and vials is Cr-Ni stainless steel. Additional operating parameters were: (a) effective milling time (10, 20, 40 and 80 h), (b) cycles frequency: 600 min^−1^, (c) 10 min on, 5 min off, in order to prevent excessive heating, (d) ball-to-power weight ratio 4.5:1 and (e) 10 grams of powder.

The morphology of the powders was followed by scanning electron microscopy (SEM, in a D500 microscope (Carl Zeiss GmbH, Jena, Germany) micrographs observation and analysis. This microscope was coupled to an EDS spectroscopy system (Bruker, Billerica, MA, USA) for the elemental microanalysis to detect atomic composition after milling. The crystallographic analysis was performed by analyzing the X-ray diffraction patterns obtained in a D500 device (Bruker, Billerica, MA, USA). The Rietveld refinement was carried out with the software MAUD (Maud, Trento, Italy). Complementary thermal analysis in order to detect thermally induced processes on heating the samples was performed in a DSC822e calorimeter (Mettler-Toledo Columbus, OH, USA). The Fe-Co-based alloys here produced are semi-hard; cycles of magnetic hysteresis, as well as susceptibility measurements, were performed in a PPMS-14T equipment with VSM and ACMS modules (Quantum Design; San Diego, CA, USA). The samples milled for 80 h were consolidated. Furthermore, some consolidated specimens were heat-treated. The coercivity of these samples was measured in a DC Foerster Koerzimat (Foerster, Reutlingen, Germany).

## 3. Results and Discussion

The microstructure evolution of the alloys during milling has been checked by X-ray diffraction. Figure 1 and Figure 2 show the diffraction patterns of alloys Co10 and Co20, respectively. Milling favors the stabilization of a bcc-Fe-rich solid solution (reflection peaks are marked with an o symbol). Furthermore, the main peak of a Nb(B) phase is detected in both alloys at intermediate milling times (10 to 40 h of milling). This phase is confirmed by Rietveld refinement (1.0 to 1.7%). The Nb(B) phase was also detected in previous works as an intermediate metastable phase [18]. At 80 h of milling, only the nanocrystalline solid solution (bcc phase) remains.

The crystallographic parameters have been calculated by Rietveld refinement (applying Maud free software, Maud, Trento, Italy). The lattice parameter, *a*, of the bcc-Fe-rich phase are given in Table 1. In both alloys, the lattice parameter, *a*, increases slightly with the milling time. This effect is due to the formation of a solid solution, with the initial addition of B and Co atoms and further addition of Nb atoms; Nb atoms probably remain in the grain boundaries due to their relatively high atomic radius. The increase is also favored by the increase of crystallographic defects. The increases were 0.59% and 0.44% for alloys Co10 and Co20, respectively. The atomic radius of Co and Fe are very similar (Fe: 0.126 nm and Co: 0.125) and the tendency of Co atoms is to remain in substitutional sites, diminishing the lattice parameter [19]. This diminution is contraposed by relatively small boron atoms in interstitial sites in the solid solution.

The parameters associated to the width of the XRD reflection (crystalline size, *L*, and microstrain, *ε*) are given in Table 2 and Table 3, respectively (before: 0 h, and after milling: 10, 20, 40 and 80 h MA).

As expected, the crystallite size decreases with the milling time. This decrease is greater in the first step of MA, from 0 to 10 h, at around 94–95% in both alloys. Further milling slightly reduces the crystallite size to ~9.5 nm, which can be considered as an asymptotic value.

The microstrain also increases with the milling time. The increase is due to the high number of crystallographic defects (as dislocations) provoked by the milling. By comparing both alloys, higher microstrain values are obtained (at all milling times) in the Co10 alloy.

Furthermore, the density of dislocation, *ρ*, (of the bcc-Fe-rich main phase) can be estimated from the crystallite size, the microstrain and the Burgers vector, *b*, taking into account the following equation [20]:*ρ* = 2 √3 (*ε/Lb*),(1)

This equation is applied to the analysis of the dislocation density of the bcc metallic phase. In this case, the direction of easy dislocations concentration is (1 1 1) and the Burgers vector is:*b* = *a* √3/2,(2)

The calculated values are given in Table 4. The dislocation density increases with the milling time and the higher values are obtained after 80 h of milling. The results of ~1 × 10^16^ m^−2^ are close to the experimental limit of the dislocation density after the plastic deformation of the metallic Fe-based alloys: ~1.310^16^ m^−2^ [21].

The morphology of MA powders was checked with electron microscopy, as shown in the Figure 3 micrographs. SEM images show a wide distribution of particles with a micrometric size and smooth and corrugated surfaces.

Furthermore, the contamination from the milling tools was checked with EDS microanalysis. After 80 h of milling, the contamination from the milling tools (Fe, Ni and Cr) is lower than 2.0 at.%. These results are similar to those previously measured in Fe-rich alloys [20,22].

Thermal analysis was carried out by DSC experiments under an Ar atmosphere. DSC scans at 20 K/min are shown in Figure 4.

In addition, the activation energy of the main crystallization process (remarked with a symbol in the DSC scans) is calculated from the Kissinger linear fitting method (linear fitting is shown in Figure 5). Activation energy is usually considered a threshold value of energy, above which any small energy fluctuation favors transformation. However, the crystallization (nucleation and/or crystal growth) of metallic alloys with several elements (such as Finemet, Nanoperm or Hitperm alloys) involve a wide variety of microstructural processes of a diverse local chemical composition. Moreover, this composition may vary over time (some atoms, such as Nb or Zr, can remain in grain boundaries), affecting kinetic parameters, such as diffusion coefficients [23]. Obviously, these complex crystallization processes mean that the activation energy is not unique (it varies with the degree of transformation). Likewise, the energies associated with nucleation and crystal growth are different. Therefore, methods such as Kissinger’s allow to obtain activation energies that are usually noted as the apparent activation energy or activation energy at the peak temperature of the transformation. One of the advantages of Kissinger’s method is that it does not presuppose a theoretical model of the analyzed process.

Regarding the analysis of both samples:(a)There are multiple exothermic processes below 500 °C that are probably associated with the structural relaxation and homogenization of alloys [16,24];(b)The main exothermic process occurs at temperatures above 500 °C and at a similar temperature interval (a slightly higher peak temperature of ~8 °C in the Co20 sample);(c)With regard to the activation energy (of the main process) calculated in the two samples with cobalt, the energy of the alloy with 10% atomic cobalt is 267 kJ/mol, whereas that of the alloy with 20% cobalt is 332 kJ/mol.

Therefore, having a higher activation energy suggests that the addition of cobalt favors the stabilization of the nanocrystalline phase obtained by mechanical grinding. This fact is confirmed because, in a previous study [24], an alloy without cobalt, Fe_85_Nb_6_B_9_, has an apparent activation energy of less than 200 kJ/mol. In addition, it is difficult to associate the activation energy values to the composition in alloys with three or more elements. The presence of minority elements in the alloy influences the activation energy. For example, the addiction of molybdenum produces an increase in the activation energy of the crystallization of the bcc phase rich in the Finemet alloys type [25]. Similarly, the addition of transition metals usually decreases activation energy [26]. On the other hand, in iron-rich alloys, the addition of cobalt is of interest because it increases the Curie temperature and, therefore, the threshold temperature (and the interval of application temperatures) in order to maintain ferromagnetic behavior.

The magnetic hysteresis loops of samples Co10 and Co20 is given in Figure 6. From a magnetic point of view, the addition of cobalt in iron-rich alloys favors an increase in the saturation magnetization [27] of the bcc-Fe-rich phase (with a maximum of approximately 20 at.% Co). It also favors an increase of the Curie temperature. In our work, the Co/Fe content ratio is 0.133 and 0.267, respectively, for alloys Co10 and Co20. The values are higher (~12 emu g^−1^) than those of alloy without Co addition [24]. This increase is also usually associated with better ordering at the atomic level (favoring magnetic interaction between Fe and Co atoms) [28]. In addition, an increase in coercivity can be produced [29], as detected in our work.

However, it is known that the addition of a metalloid element does not favor high values in the saturation magnetization. This is caused by a portion of the electrons in the 3D outer layer of the ferromagnetic elements and is linked to atoms of the non-magnetic element, decreasing the total number (and density) of electrons of the 3D outer layer that provide effective magnetic moment to the sample. The specific effect depends on the non-metallic element. For example, phosphorus addition causes a greater reduction in saturation magnetization than boron addition [30]. The results of Table 5 show that coercivity effectively increases with the partial replacement of Fe by Co. Instead, the magnetization remains virtually constant. This fact (being similar to the crystalline sizes in both compositions) is probably caused by a higher rate of microstrain in the Co10 sample, since crystallographic defects hinder both the movement and the ordering of magnetic domains by increasing the saturation magnetization.

Complementary magnetic behavior is given by the remanence (or remanent field) *M_r_* and the squareness ratio (*M_r_/M_s_*). For mechanical alloying, the value of *M_r_/M_s_* is usually low, at between 0.01 and 0.1 [31]. A value close to 1 is suitable for magnetic storage systems. On the other hand, values close to zero are suitable for transformer cores. The values of the magnetic parameters are given in Table 5.

Magnetic susceptibility in frequency measurements (AC magnetic susceptibility) has two components (real and imaginary). The real component measures the degree of magnetization of a material in response to the applied magnetic field. The imaginary component is related to the outdatedness between the magnetic response (due to the magnetic energy loss) of the material and the magnetic field applied. Generally, if the main phase does not vary, regarding the mechanical milling, the values of both components increases with the grinding milling [32]. In this study, we characterize the samples milled for 80 h by applying an external magnetic field with a low value (1 Oe). The frequencies chosen were 333, 666 and 110 Hz. In both samples, measurements have been made in ZFC (zero field cooling) conditions, corresponding to the warming of the sample by applying the field when the sample has previously been cooled without applying the field. In the case of the Co20 sample, there is also the measurement in FC (field cooling) conditions, corresponding to a later cooling applying field.

The real component in Figure 7 shows that, at the studied frequencies, there is a tendency to decrease, especially from 200 K, without reaching a value close to zero that would indicate the Curie temperature of the sample. The ZFC and FC curves of the Co10 sample are different (especially at a frequency of 333 Hz). In this case the difference increases by decreasing the temperature almost continuously. This may be related to the fact that, at low temperatures, there are regions with superparamagnetic or spin glass behavior. Both behaviors have been detected in samples obtained by mechanical milling [33].

At a fixed frequency, the values of the imaginary component (including the FC measurements) remain constant up to 300 K, as shown in Figure 8. This indicates that there are no significant structural or magnetic transformations, including the magnetic relaxation processes, that are typical of nanostructured samples and nanoparticles [34]. Consequently, small fluctuations in both signals detected may be associated with inhomogeneities in the atomic-level magnetic ordering of Fe and Co atoms.

Regarding the influence of specific frequencies, a clear trend is not detected. Low working frequencies (<5000 Hz, far from the typical frequencies of ferromagnetic resonance, GHz) favor a non-stable response of the susceptibility versus frequency [35].

The next step following the mechanical alloying process is usually powder compaction. One of the main technological issues is to preserve/obtain the desired microstructure; in this case, to preserve their non-equilibrium nanocrystalline structure. Various techniques are employed for powder consolidation, such as cold pressing followed by sintering, hot pressing, hot isostatic pressing, explosive compaction and spark plasma sintering [36]. In many cases, powder compaction is an alternative to the rapid quenching (casting) technique, which can obtain bulk amorphous alloys and their composites. The compaction of mechanically alloyed powders/composites can also help to overcome many problems related to the traditional manufacturing of metal matrix composites, such as particle agglomeration, low wettability or interfacial reactions.

As the produced alloys obtained are magnetically semi-hard, for its applicability, it is necessary that this magnetic response remains after the processes of consolidation and/or sintering. The alloys Co10 and Co20 that were compacted after 80 h of milling were analyzed. The specimens were made by pressing at 600 MPa for 30 min in vacuum. The disks dimensions were 10mm in diameter and approximately 3 mm thick. The temperatures selected for annealing treatments were room temperatures of 350 and 600 °C. Six samples of each composition were prepared (two untreated, two annealed at 350 °C and one annealed at 600 °C). Measurements were performed in open magnetic circuit by applying OIEC 60404-7 method B (in order to measure test specimens that are almost geometry-independent).

The Table 6 shows the coercivity values of the associated average (six measurements in two samples of the same composition) and error. The results are compared with those of specimens with similar composition.

Regarding the influence of the partial replacement of Fe by Co., cobalt, at 10 at-%, causes a decrease (and at 20 at.%, causes an increase) in coercivity, whereas Ni addition always causes a diminution. Certainly, mechanical alloying provokes a high density of crystallographic defects [37].

Heat treatment at low temperatures helps to reduce the density of crystallographic defects (including dislocations) by favoring both diffusion and a reduction of the density of crystallographic defects [37,38]. It is known that treatments at low temperatures (around 300–400 °C) cause a relaxation of structural tensions, a slight increase in saturation magnetization and a decrease in coercivity [38]. A high annealing temperature usually induces the crystalline growth of the sample and leads to crystallization in amorphous alloys or an increase in the crystalline size in nanocrystalline alloys (with a consequent increase in coercivity) [38,39]. This effect is here detected when performing annealing at 600 °C in all samples. Thus, annealing at 350 °C reduces coercivity, whereas annealing at 600 °C provokes a contrary response (an increase in coercivity) due to the crystalline growth detected by calorimetry (the main exothermic process with a peak shape).

## 4. Conclusions

Nanocrystalline Fe_75_Co_10_Nb_6_B_9_ and Fe_65_Co_20_Nb_6_B_9_ have been prepared by mechanical alloying. After 80 h of milling, only the nanocrystalline Fe-rich solid solution (bcc phase) remains. The crystallite size is ~9.5 nm.

Multiple exothermic processes below 500 °C (that are probably associated with structural relaxation and alloys homogenization) were found by calorimetry. The activation energy (of the process near 600 °C) was calculated in the two samples: the energy of the alloy with 10% atomic cobalt is 267 kJ/mol, whereas that of the alloy with 20% cobalt is 332 kJ/mol.

From the magnetic analysis, the addition of cobalt in iron-rich alloys provokes a variation in saturation magnetization and coercivity. Likewise, the coercivity of pressed specimens allows us to state that annealing at 350 °C improve the coercivity reduction (probably by reducing the internal microstrain), whereas annealing at 600 °C reduces this effect (probably due to the crystalline growth).

## Figures and Tables

**Figure 1 materials-14-04542-f001:**
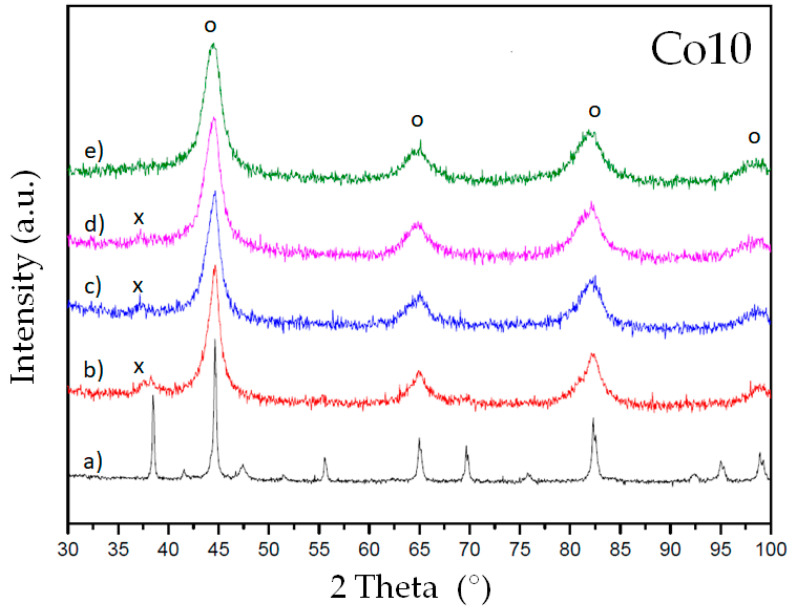
XRD diffraction patterns of alloy Co10: (**a**) before milling, (**b**) 10 h milling, (**c**) 20 h milling, (**d**) 40 h milling and (**e**) 80 h milling. Symbols: o marks the bcc reflections of the main phase and x marks the main reflection of a Nb(B) phase.

**Figure 2 materials-14-04542-f002:**
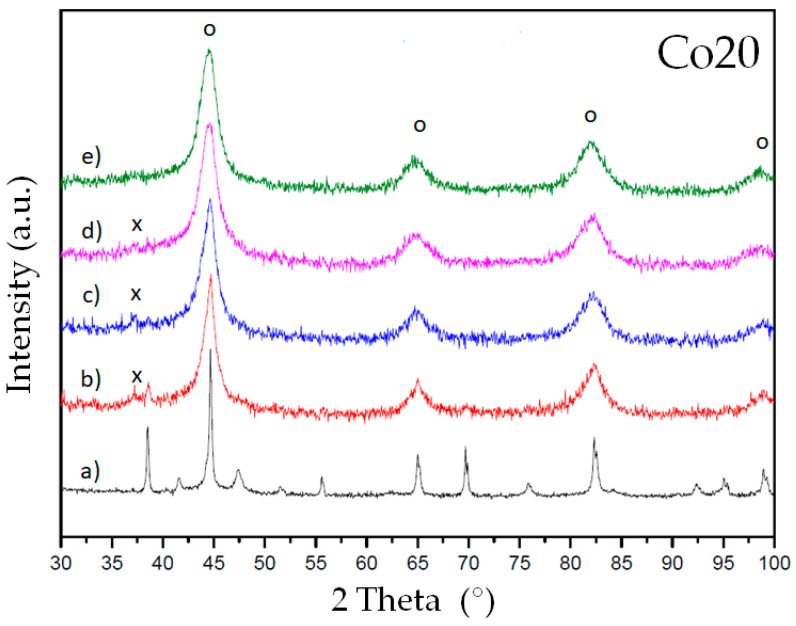
XRD diffraction patterns of alloy Co20: (**a**) before milling, (**b**) 10 h milling, (**c**) 20 h milling, (**d**) 40h milling and (**e**) 80 h milling. Symbols: o marks the bcc reflections of the main phase and x marks the main reflection of a Nb(B) phase.

**Figure 3 materials-14-04542-f003:**
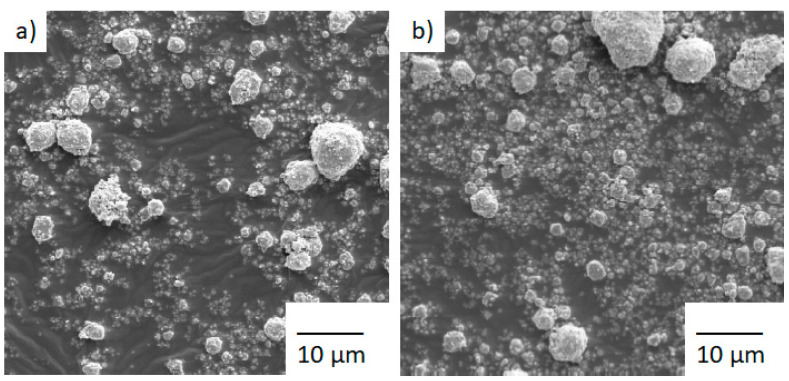
SEM micrographs of particles of alloys Co10 ((**a**) left) and Co20 ((**b**) right) milled for 80 h.

**Figure 4 materials-14-04542-f004:**
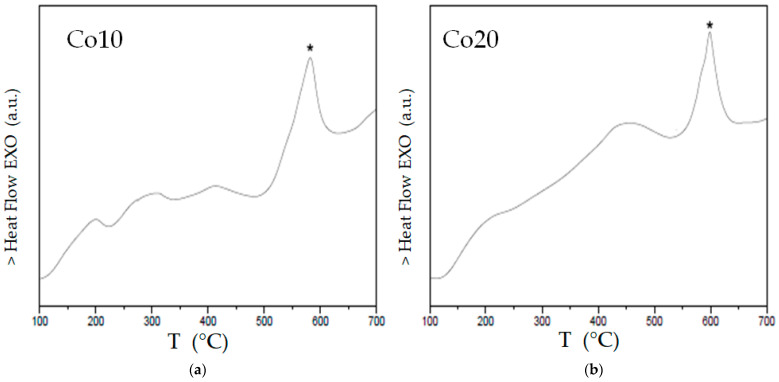
DSC scans (20 K/min) of alloys Co10 ((**a**) left) and Co20 ((**b**) right) milled for 80 h. Symbol: * marks the peak of the main crystallization peak.

**Figure 5 materials-14-04542-f005:**
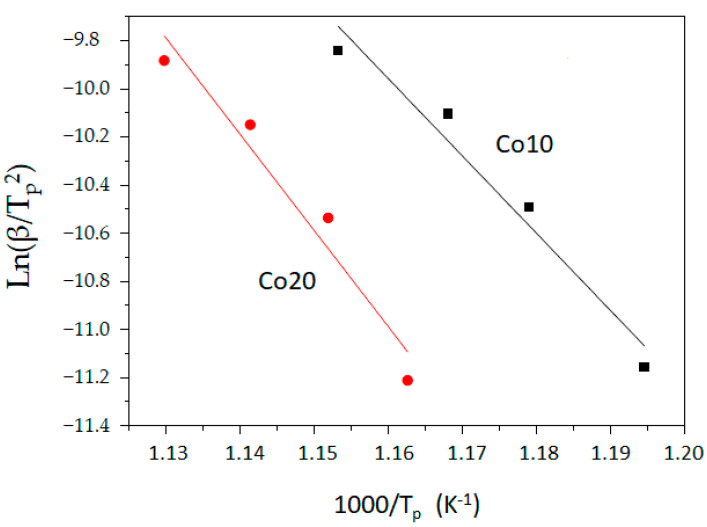
Kissinger plots from the DSC scans at different heating rates of alloys Co10 (■ right) and Co20 (● left).

**Figure 6 materials-14-04542-f006:**
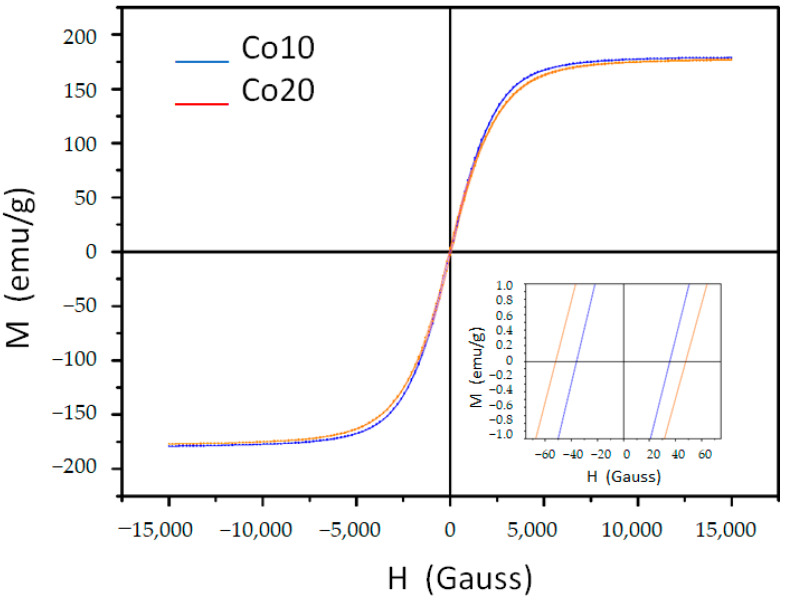
Magnetic hysteresis loops of both alloys milled for 80 h. The inset corresponds to the (0, 0) area.

**Figure 7 materials-14-04542-f007:**
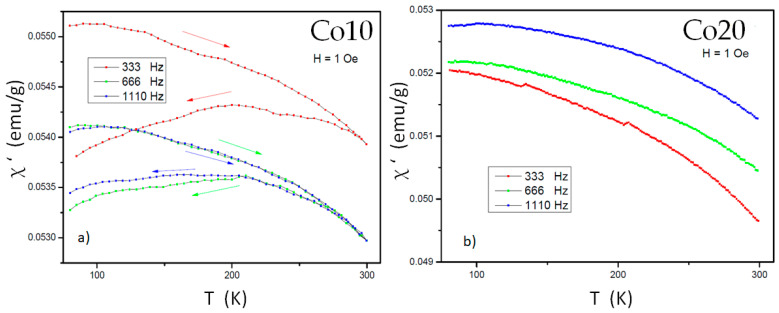
Magnetic susceptibility (real component) of both alloys milled for 80 h (at different frequencies: (**a**) alloy Co10, (**b**) alloy Co20.

**Figure 8 materials-14-04542-f008:**
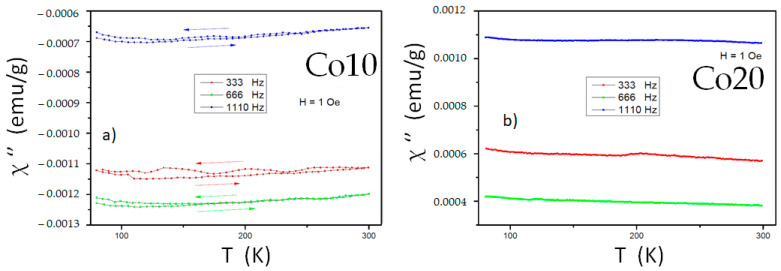
Magnetic susceptibility (imaginary component) of both alloys milled for 80 h (at different frequencies): (**a**) alloy Co10, (**b**) alloy Co20.

**Table 1 materials-14-04542-t001:** Lattice parameter, *a* (Å), of the iron-rich bcc phase as a function of milling time.

Alloy	0 h	10 h	20 h	40 h	80 h
Co10	2.8666 ± 0.0004	2.8736 ± 0.0003	2.8797 ± 0.0004	2.8825 ± 0.0003	2.8836 ± 0.0003
Co20	2.8665 ± 0.0005	2.8717 ± 0.0003	2.8758 ± 0.0004	2.8794 ± 0.0004	2.8791 ± 0.0003

**Table 2 materials-14-04542-t002:** Crystalline size, *L* (nm), of the iron-rich bcc phase as a function of milling time.

Alloy	0 h	10 h	20 h	40h	80 h
Co10	212 ± 6	12.9 ± 0.2	11.9 ± 0.3	9.8 ± 0.2	9.5 ± 0.3
Co20	206 ± 6	11.3 ± 0.2	9.5 ± 0.2	8.0 ± 0.3	9.8 ± 0.4

**Table 3 materials-14-04542-t003:** Microstrain, *ε* (%), of the iron-rich bcc phase as a function of milling time.

Alloy	0 h	10 h	20 h	40 h	80 h
Co10	0.013 ± 0.003	0.35 ± 0.02	0.58 ± 0.08	0.62 ± 0.08	0.70 ± 0.05
Co20	0.010 ± 0.003	0.11 ± 0.02	0.32 ± 0.06	0.43 ± 0.05	0.59 ± 0.03

**Table 4 materials-14-04542-t004:** Dislocation density, *ρ* (units: 10^15^ m^−2^), of the iron-rich bcc phase as a function of milling time.

Alloy	0 h	10 h	20 h	40 h	80 h
Co10	0.016 ± 0.005	4.32 ± 0.05	7.8 ± 0.5	10.0 ± 0.6	12 ± 2
Co20	0.008 ± 0.009	1.0 ± 0.1	5.8 ± 0.6	8.7 ± 0.5	10 ± 2

**Table 5 materials-14-04542-t005:** Magnetic parameters: saturation magnetization, *M_s_*, coercivity, *H_c_*, remanence, *M_r,_*, and squareness ratio, *M_r_/M_s_*.

Sample	*M_s_*/emu g^−1^	*H_c_*/Oe	*M_r_*/emu g^−1^	*M_r_/M_s_*/Adim.
Co10	178.9	36	2.52	0.014
Co20	177.0	50	3.38	0.019

**Table 6 materials-14-04542-t006:** Coercivity, *H_c_*, of compacted specimens at room temperature (RT) and after annealing (30 min) at 350 or 600 °C.

Sample	*H_c_* (RT)/A m^−1^	*H_c_* (350 °C)/A m^−1^	*H_c_* (600 °C)/A m^−1^
Fe_85_Nb_6_B_9_	2747 ± 29	1868 ± 41	6145 ± 15
Co10(Fe_75_Co_10_Nb_6_B_9_)	2054 ± 85	1786 ± 45	5695 ± 15
Co20(Fe_65_Co_20_Nb_6_B_9_)	3211 ± 37	2326 ± 44	4734 ± 72
Fe_75_Ni_10_Nb_6_B_9_	2015 ± 28	1883 ± 15	3826 ± 35
Fe_65_Ni_120_Nb_6_B_9_	1859 ± 27	1559 ± 31	2582 ± 66

## Data Availability

Data can be requested to the authors.

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
