# Peer review of "Structural, Thermal and Magnetic Analysis of Fe75Co10Nb6B9 and Fe65Co20Nb6B9 Nanostructured Alloys"

_materials, 2021, doi:10.3390/ma14164542_

Round 1

Reviewer 1 Report

In the submitted manuscript you described the preparation of Fe-Co-Nb-B alloys with different Co content. 

I find the research interesting, but there are some issues that has to be tackled prior to the publishing. 

  • ball-to-powder ratio is missing
  • data on the equipment has to be checked. I highly doubt that all the equipment is from the same producer
  • line 80: unclear, check english
  • line 114: 0 h (not oh)
  • comment on the microstrain in Co20 is missing
  • line 135: "electronic microscopy" - use the correct terminology
  • SEM analysis and size distribution: from the presented images one can not see what you are describing that one should see - agglomerates and individual particles can be seen, but not the size distribution.
  • how the particles' size was determined, which method was used and on how many particles the analysis was executed? Please add to the manuscript. 
  • line 171: "grain borders" - use appropriate terminology
  • line 201: unclear, looks like a sentence is missing
  • line 205: uclear 
  • l.208-211: unclear 
  • l. 213: unclear
  • l. 235-236: unclear
  • l.274: uclear
  • How the Hc of bulk pieces was determined? Please add to the manuscript
  • l.285: According to the Table 7, this statement is not correct
  • l.290-294: not supported by the presented results

Author Response

Reviewer 1:

In the submitted manuscript you described the preparation of Fe-Co-Nb-B alloys with different Co content. 

I find the research interesting, but there are some issues that has to be tackled prior to the publishing. 

  • ball-to-powder ratio is missing

Answer: We add the ball to powder weight ratio:

  1. d) ball-to-power weight ratio 4.5:1 and …
  • data on the equipment has to be checked. I highly doubt that all the equipment is from the same producer

Answer: We check the producer information with the information given by the Research Services Facilities of our university and we modify this information in the manuscript.

 The morphology and size distribution of the powders has been followed by scanning electron microscopy (SEM, in a D500 microscope (Carl Zeiss GmbH, Jena, Germany) micrographs observation and analysis. This microscope is coupled to an EDS spectroscopy system (Bruker, Billerica, MA, USA) for the elemental microanalysis to detect atomic composition after milling.  The crystallographic analysis was performed by analyzing the X-Ray diffraction patterns obtained in a D500 device (Bruker, Billerica, MA, USA).

Serveis Tècnics de Recerca > Aparells / tècniques disponibles > Anàlisi Química i Estructural > Difracció de raigs X de pols (XRPD) (udg.edu)

Serveis Tècnics de Recerca > Aparells / tècniques disponibles > Microscòpia i Biotecnologia > Microscòpia electrònica de rastreig (MER) (udg.edu)

  • line 80: unclear, check english

Answer: We clarify this sentence. It’s true, it was unclear.

  • The samples milled 80 hours were consolidated. Furthermore, some consolidated specimens were heat-treated. The coercivity of these samples was measured in a DC Foerster Koerzimat (Foerster, Reutlingen, Germany).
  • line 114: 0 h (not oh)

Answer: We agree this correction. In our version this error corresponding to line 112. Probably, there is a shift due to word versions. This shift can affect next answers to your comments.

(before: 0 h, and after milling: 10, 20, 40 and 80 h MA).

  • comment on the microstrain in Co20 is missing

Answer: We modify the sentence to take into account Co20 sample microstrain evolution.

By comparing both alloys, higher microstrain values are obtained (at all milling times) in Co10 alloy.

  • line 135: "electronic microscopy" - use the correct terminology

Answer: We agree this correction. In our version this error corresponding to line 130. Probably, there is a shift due to word versions. This shift can affect next answers to your comments. We modify this expression.

The morphology of MA powders was checked with electron microscopy,

  • SEM analysis and size distribution: from the presented images one can not see what you are describing that one should see - agglomerates and individual particles can be seen, but not the size distribution.

Answer: We amplify some areas and use several micrographs- The applied method to determine the particle size is given in our answer to the next comment.

  • how the particles' size was determined, which method was used and on how many particles the analysis was executed? Please add to the manuscript.

Answer: The method is now commented in the manuscript. I also provide reviewer an example.

The particle diameter has been calculated as the semi sum of the two high perpendicular high diameters of the particle.

  • line 171: "grain borders" - use appropriate terminology

Answer: We modify the terminology.

(some atoms such as Nb or Zr can remain in grain boundaries)

  • line 201: unclear, looks like a sentence is missing
  • line 205: uclear 
  • l.208-211: unclear 
  • l. 213: nuclear

Answer: we introduce some changes in this part of the manuscript to clarify meaning.

This increase is also usually associated with better ordering at the atomic level (favoring magnetic interaction between Fe and Co atoms) [28]. In addition, it can be produced an increase in coercivity [29], as detected in our work.

However, it is known that the addition of a metalloid element does not favor high values in saturation magnetization. This is caused by a portion of the electrons in the outer layer 3d of the ferromagnetic elements and is linked to atoms of the non-magnetic element.; decreasing the total number (and density) of electrons of the 3d outer layer that provide effective magnetic moment to the sample. The specific effect depends on the non-metallic element. For example, phosphorus addition causes a greater reduction in saturation magnetization than boron addition [30]. The results of table 6. they show that coercitivity effectively increases with the partial replacement of Fe by Co. Instead, magnetization remains virtually constant. This fact (being similar the crystalline sizes in both compositions) is probably caused by a higher rate of microstrain in the Co10 sample, since crystallographic defects hinder both the movement and the ordering of magnetic domains by increasing the saturation magnetization.

Complementary magnetic behavior is given by the remanence (or remanent field) Mr and the squareness ratio (Mr/Ms). For mechanical alloying the value of Mr/Ms is usually low, between 0.01 and 0.1 [31]. A value close to 1 is suitable for magnetic storage systems. On the other hand, values close to zero are suitable for transformer cores. The values of the magnetic parameters are given in the table 6.

  • l. 235-236: nuclear

Answer: We modify some sentences to clarify this part of the manuscript.

In this study, we characterize the samples milled for 80 hours by …

. This may be related to the fact that at low temperatures there are regions with superparamagnetic or spin glass behavior. Both behaviors have been detected in samples obtained by mechanical milling [33].

  • l.274: unclear

Answer: We modify some sentences to clarify this part of the manuscript.

  • Regarding the influence of partial replacement of Fe by Co., cobalt causes at 10 at-% an decrease (and at 20 at.% an increase) in coercivity, whereas Ni addition always a diminution. Certainly, mechanical alloying provokes a high density of crystallographic defects [37]. Heat treatment at low temperatures helps to reduce the density of crystallographic defects (including dislocations) by favoring diffusion and a reduction of the density of crystallographic defects [37, 38]. It is known that treatments at low temperatures (around 300-400 °C) cause a relaxation of structural tensions, a slight increase in saturation magnetization and a decrease in coercivity [38]. A high annealing temperature usually induces the crystalline growth of the sample and leads to the crystallization in amorphous alloys or the increase of the crystalline size in nanocrystalline alloys (with the consequent increase of coercivity) [38, 39]. This effect is here detected when doing annealing at 600 °C in all samples. Thus, annealing at 350°C reduces coercivity, whereas annealing at 600°C provokes contrary response (coercivity increase) due to the crystalline growth detected by calorimetry (main exothermic process with peak shape).
  • How the Hc of bulk pieces was determined? Please add to the manuscript

Answer: We add this information to the manuscript.

Measurements were performed in open magnetic circuit by applying OIEC 60404-7 method B (to measure test specimens almost geometry independent).

  • l.285: According to the Table 7, this statement is not correct

Answer: We agree this comment. We modify this statement.

Regarding the influence of partial replacement of Fe by Co., cobalt causes at 10 at-% an decrease (and at 20 at.% an increase) in coercivity, whereas Ni addition always a diminution.

  • l.290-294: not supported by the presented results

Answer: We modify this part. The introduce “probably” and we delete a sentence.

The new version is.

Multiple exothermic processes below 500°C (probably associated with structural relaxation and alloys homogenization) were found by calorimetry. The activation energy (of the process near 600°C) was calculated in the two samples, the energy of the alloy with 10% atomic cobalt is 267 kJ/mol while that of the alloy with 20% cobalt is 332 kJ/mol.

Reviewer 2 Report

Average paper.

There are many language errors, e.g.

  1. “magnetization of saturation” – should be: saturation magnetization
  2. Line 77 “..heating the samples was done performed in …” – syntax is wrong
  3. Line 81 “… samples milled 80 hours milled samples consolidated and …” – lack of predicate
  4. Many sentences start with “likewise”

The text is written untidy and should be read carefully once again.

Merit remarks:

  1. There is no confirmation that the XRD peak detected in both alloys at intermediate milling times (10-40 hours) belongs to Nb(B) phase.
  2. What does it mean that contamination is below 2 at. %? Contamination by Cr and Ni only (coming from Cr-Ni steel used as milling device) or Fe is also considered?
  3. Looking at the DSC curves is difficult to estimate the baseline position? Maybe the baseline (second heating) was subtracted already?  The authors claim that the observed transformations can involve a wide variety of microstructural processes (relaxation, homogenization), however, this issue is not explained. XRD studies of the samples heated just below the main crystallization peak would be useful to clarify the issue.  
  4. The values of coercivity presented in table 6 (36 and 50 Oe for Co10 and Co20, respectively), correspond to 2880 and 4000 A/m, therefore, the statement “as the produced alloys obtained are magnetically soft” (line 274) is not true! It is commonly accepted that coercivity above 1000 A/m (1 kA/m) corresponds to semi-hard magnets.
  5. Surprisingly, in Table 7 the coercivity of the same (?) samples before annealing is 2054 and 3211 A/m. Which values are true?
  6. Moreover, the coercivity of the samples subjected to pressing and annealing at 350 C decreases comparing to milled samples only, while heat treatment at 600 C results in the increase of coercivity. What is the reason for such behavior? XRD pattern of the heat-treated samples should be included in the paper.
  7. In the case of milled samples, the magnetic properties were measured for loose powder, while for heat-treated samples – the powders were pressed (600 MPa) into discs. Therefore, the type of sample can influence the measured magnetic properties.

Author Response

Reviewer 2:

Average paper.

There are many language errors, e.g.

  1. “magnetization of saturation” – should be: saturation magnetization

Answer: We modify this expression. It has been performed in all the sentences of the manuscript with this term. As an example:

Magnetic parameters: Saturation magnetization, Ms, coercivity, Hc, remanence, Mr, and squareness ratio, Mr/Ms.

  1. Line 77 “..heating the samples was done performed in …” – syntax is wrong

Answer: We agree this comment. We modify.

Complementary thermal analysis to detect thermally induced processes on heating the samples was performed in …

  1. Line 81 “… samples milled 80 hours milled samples consolidated and …” – lack of predicate

Answer: We modify this sentence.

The samples milled 80 hours were consolidated. Furthermore, some consolidated specimens were heat-treated.

  1. Many sentences start with “likewise”

Answer: We check the manuscript. In the revised version, likewise only appear two. We use as alternatives the following terms-

In addition, Similarly, Moreover, Besides.

The text is written untidy and should be read carefully once again.

Answer: We read gain the manuscript. More than 20 modifications were introduced in the new version of the manuscript. Some sentences were unclear. These changes are remarked in the manuscript. Some examples are here given.

before: 0 h,

By comparing both alloys, higher microstrain values are obtained (at all milling times) in Co10 alloy.

electron microscopy

grain boundaries)

This increase is also usually associated with better ordering at the atomic level (favoring magnetic interaction between Fe and Co atoms) [28].

Merit remarks:

  1. There is no confirmation that the XRD peak detected in both alloys at intermediate milling times (10-40 hours) belongs to Nb(B) phase.

Answer: Rietveld refinement was performed by adding this phase. Minor amount was found. We add this information to the manuscript.

This phase is confirmed by Rietveld refinement (1.0 to 1.7%).

  1. What does it mean that contamination is below 2 at. %? Contamination by Cr and Ni only (coming from Cr-Ni steel used as milling device) or Fe is also considered?

Answer: We consider Cr, Ni and Fe. We clarify this query in the new version.

After 80 hours of milling the contamination from the milling tools (Fe, Ni Cr) is lower than 2.0 at.%.

  1. Looking at the DSC curves is difficult to estimate the baseline position? Maybe the baseline (second heating) was subtracted already?  The authors claim that the observed transformations can involve a wide variety of microstructural processes (relaxation, homogenization), however, this issue is not explained. XRD studies of the samples heated just below the main crystallization peak would be useful to clarify the issue.

Answer: Yes, the baseline (second heating) was substracted. We add this information in the manuscript. Regarding additional XRD confirmation, it has not been performed. Nevertheless, two references has been used to collaborate these comments. We also add “probably”

  1. There are multiple exothermic processes below 500°C that are probably associated with structural relaxation and homogenization of alloys [16, 24].
  2. The values of coercivity presented in table 6 (36 and 50 Oe for Co10 and Co20, respectively), correspond to 2880 and 4000 A/m, therefore, the statement “as the produced alloys obtained are magnetically soft” (line 274) is not true! It is commonly accepted that coercivity above 1000 A/m (1 kA/m) corresponds to semi-hard magnets.

Answer: It is true. Some authors consider our values as soft magnetic, but it should be considered as semi-hard. We adapt the manuscript using “semi-hard” and expressions as “reducing the coercivity” or “inhibiting the …”. The changes were remarked in the manuscript. Some of them are.

inhibiting the coercivity increase

As the produced alloys obtained are magnetically semi-hard.

Thus, annealing at 350°C reduces coercivity, whereas

improve coercivity reduction

  1. Surprisingly, in Table 7 the coercivity of the same (?) samples before annealing is 2054 and 3211 A/m. Which values are true?

Answer: It was our mistake. Is not the same sample. There are consolidated samples Co10 and Co20. We correct information about sample in table 7.

Co20(Fe65Co20Nb6B9)

  1. Moreover, the coercivity of the samples subjected to pressing and annealing at 350 C decreases comparing to milled samples only, while heat treatment at 600 C results in the increase of coercivity. What is the reason for such behavior? XRD pattern of the heat-treated samples should be included in the paper.

Answer: Crystallographic defects provokes an increase of the coercivity. Annealing at low temperatures favors diffusion, the reduction of the crystallographic defects and a reduction of coercivity. Annealing at 600ºC favors the crystalline growth. Some sentences have been modified in this paragraph and a new reference was added.

Certainly, mechanical alloying provokes a high density of crystallographic defects [37]. Heat treatment at low temperatures helps to reduce the density of crystallographic defects (including dislocations) by favoring diffusion and a reduction of the density of crystallographic defects [37, 38]. It is known that treatments at low temperatures (around 300-400 °C) cause a relaxation of structural tensions, a slight increase in saturation magnetization and a decrease in coercivity [38]. A high annealing temperature usually induces the crystalline growth of the sample and leads to the crystallization in amorphous alloys or the increase of the crystalline size in nanocrystalline alloys (with the consequent increase of coercivity) [38, 39]. This effect is here detected when doing annealing at 600 °C in all samples. Thus, annealing at 350°C reduces coercivity, whereas annealing at 600°C provokes contrary response (coercivity increase) due to the crystalline growth detected by calorimetry (main exothermic process with peak shape).

  1. Alleg, S., Souilah, S., Younes, A., Bensalem, R., Suñol, J.J., Greneche, J.M. Effect of the Nb content on the amorphization process of the mechanically alloyed Fe-Co-Nb-B powders. Alloys & Comp. 2012, 5365, 394-397.

  1. In the case of milled samples, the magnetic properties were measured for loose powder, while for heat-treated samples – the powders were pressed (600 MPa) into discs. Therefore, the type of sample can influence the measured magnetic properties.

Answer: The type of sample as well as the compaction conditions can influence the magnetic properties. In this work, we measure (table 7) we also measure the pressed specimens before annealing, We clarify in the text and in the table caption.

Six samples of each composition of have been prepared (two un-treated, two annealed at 350°C and annealed at 600°C).

Table 7. Coercivity, Hc, of compacted specimens at room temperature (RT) and after annealing (30 minutes) at 350 or 600°C.

Reviewer 3 Report

The submitted manuscript is entitled ‘Structural, thermal and magnetic analysis of Fe75Co10Nb6B9 and Fe65Co20Nb6B9 nanostructured alloys”. The structure of the scientific report is excellent and well-understood. The aim is clarified. The introduction summarizes to fill this literature gap. Bibliographic references are appropriate. The author mentioned 38 literature according to the reference list. These references are from the last 10 years, from prestigious journals.

It is often used to mark images of the right and left marks. Please use instead marking a) and b).

In Figure 6, the left and right markings are missing.

The reviewer suggests accepting after minor revision form in the paper for publication.

Author Response

Reviewer 3:

The submitted manuscript is entitled ‘Structural, thermal and magnetic analysis of Fe75Co10Nb6B9 and Fe65Co20Nb6B9 nanostructured alloys”. The structure of the scientific report is excellent and well-understood. The aim is clarified. The introduction summarizes to fill this literature gap. Bibliographic references are appropriate. The author mentioned 38 literature according to the reference list. These references are from the last 10 years, from prestigious journals.

It is often used to mark images of the right and left marks. Please use instead marking a) and b).

Answer: We add these marks in the figures and also in the figure captions. This procedure has been applied to figures 3, 4, 5, 8 and 9. As examples, we add here new figure 3 and the figure captions.

Figure 3. SEM micrographs of particles of alloys Co10 (a: left) and Co20 (b: right) milled for 80 hours.

Figure 4. Histograms with particles size distribution of alloys Co10 (a: left) and Co20 (b: right) milled for 80 hours.

Figure 5. DSC scans (20 K/min) of alloys Co10 (a: left) and Co20 (b: right) milled for 80 hours. Symbol: * mark the peak of the main crystallization peak.

Figure 8. Magnetic susceptibility (real component) of both alloys milled for 80 hours (at different frequencies. a) Alloy Co10, b) Alloy Co20.

Figure 9. Magnetic susceptibility (imaginary component) of both alloys milled for 80 hours (at different frequencies). a) Alloy Co10, b) Alloy Co20.

In Figure 6, the left and right markings are missing.

Answer: We add left and right markings in the figure caption of figure 6.

Figure 6. Kissinger plots from the DSC scans at different heating rates of alloys Co10 (▪ right) and Co20 (● left).

The reviewer suggests accepting after minor revision form in the paper for publication.

Round 2

Reviewer 1 Report

Dear authors, 

You have made significant improvements of your manuscript. I have only one comment, the particles' size. From what you described as your method I can not accept the revised manuscript as it is. Do the particles' size analysis as it should be done (if you don't experience, ask someone who has them). Moreover, what you presented in the additional Figure following your answer, it can be clearly seen, that the "particle" you were measuring is an agglomerate of smaller particles. Also information on how many particles the analysis was done, is missing. 

Author Response

Answer:

We agree the comment of the reviewer. Our method can be considered as an Average diameter method. There are methods to determine the particle size Distribution: by sieving, with software as Image J or by experimental particle size analysis in particle sizer devices.

The example we provide with a conglomerate was to clearly show the procedure. For the analysis big particles were not taken into account and five micrographs of each sample were analyzed. In order to send quickly a revised version of the manuscript (August is our summer holiday period) one option is to replace Particle size by Average diameter. Nevertheless, as the information given in this figure is not significative in our study the best option is to delete the figure and the paragraph. Thus, in the new version of the manuscript we delete both.